# Polymer Translocation Across a Corrugated Channel: Fick–Jacobs Approximation Extended Beyond the Mean First-Passage Time

**DOI:** 10.3390/polym11020251

**Published:** 2019-02-02

**Authors:** Paolo Malgaretti, Gleb Oshanin

**Affiliations:** 1Max-Planck-Institut für Intelligente Systeme, Heisenbergstr. 3, D-70569 Stuttgart, Germany; 2IV. Institut für Theoretische Physik, Universität Stuttgart, Pfaffenwaldring 57, D-70569 Stuttgart, Germany; 3Sorbonne Université, CNRS, Laboratoire de Physique Théorique de la Matière Condensée, LPTMC (UMR CNRS 7600), 4 Place Jussieu, CEDEX 05, 75252 Paris, France; oshanin@lptmc.jussieu.fr

**Keywords:** polymer translocation, first-passage time, entropic barrier

## Abstract

Polymer translocation across a corrugated channel is a paradigmatic stochastic process encountered in diverse systems. The instance of time when a polymer first arrives to some prescribed location defines an important characteristic time-scale for various phenomena, which are triggered or controlled by such an event. Here we discuss the translocation dynamics of a Gaussian polymer in a periodically-corrugated channel using an appropriately generalized Fick–Jacobs approach. Our main aim is to probe an effective broadness of the first-passage time distribution (FPTD), by determining the so-called coefficient of variation γ of the FPTD, defined as the ratio of the standard deviation versus the mean first-passage time (MFPT). We present a systematic analysis of γ as a function of a variety of system’s parameters. We show that γ never significantly drops below 1 and, in fact, can attain very large values, implying that the MFPT alone cannot characterize the first-passage statistics of the translocation process exhaustively well.

## 1. Introduction

Transport of polymers across pores or channels is a crucial process for many biological, physical, as well as for some man-made systems (see, e.g., Refs. [1,2,3]). For example, a viral infection is triggered by the injection of a viral DNA/RNA into the cell through a narrow pore that links the viral capsid to the cell membrane [4,5]. The cell cycle is controlled by proteins, that are synthesized in a cell nucleus and need to translocate across the nuclear membrane [6]. A gel electrophoresis—an efficient polymer separation technique, takes advantage of the strong dependence of the polymer mobility on its length, acquired due to the transport of the latter across a gel with an artificially prepared or a naturally occurring highly-porous structure [7,8]. This permits us to separate the polymers in a rather robust way with respect to their molecular weight. More recently, nanofluidic channels with a specially designed geometry have been used to manipulate single polymer chains [9]. These systems function at minute, nano-molar concentrations of polymers, so that the net outcome, e.g., a viral infection, a cell regulation and a polymer translocation across a porous medium, is controlled by the first-passage event of a single molecule, rather than by a steady molecular current across the system. As a consequence, here the relevant property is the first-passage time distribution (FPTD) [10,11]—a distribution of the instants of time when a polymer, released at some fixed point, arrives to a prescribed location for the first time.

The analysis of the first-passage phenomena concentrates traditionally on the first moment of the FPTD—the so-called mean first-passage time (MFPT)—the inverse of the mean translocation rate [1,2,3,12,13,14,15,16]. However, it is not clear a priori to which extent the first moment of the distribution, which is expected to have a complex structure, characterizes the dynamical behavior in the system exhaustively well. Nonetheless, it is often taken for granted that the FPTD is sufficiently narrow and the only pertinent temporal scale is the MFPT. In some particular cases this may be true, but one cannot expect that it holds in general. In particular, it is well-documented that even in the simplest settings and in bounded systems (where all the moments of the FPTD exist), first-passage events take place with a great disparity in time [17,18,19], such that the MFPT can differ from the typical, most probable first-passage times by orders of magnitude [20,21]. Such a disparity becomes progressively more pronounced and the FPTD becomes effectively more broad, the closer the starting point to the target is [17,18,19,20,21]—there are a few “direct” trajectories arriving to the target in a very short time [22], (which defines the most probable time), but most of trajectories go away from the target and sample the whole volume before they eventually return to the target and react. Quite generally, for small statistical samples, (i.e., when one garners only few realizations of trajectories), it is most likely to observe the first-passage times which are substantially less than the MFPT. It is then not a big surprise that the latter often turns out to be supported by atypically long trajectories and hence, is associated with the tails of the FPTD [20,21]. As a consequence, calculation of the MFPT in numerical or experimental analyses usually necessitates an averaging over a large statistical sample, with rare, anomalously long trajectories providing a significant input to the average value. One should therefore be rather cautious when using the available analytical expressions for the MFPT in order to explain the kinetic behavior in systems in which only a single or a few molecules are present [20,21]. We also note that similar effects and a wide distribution of the translocation times have been observed for driven polymer transport in non-corrugated pores, both in experiments [23,24] and in numerical simulations (see, e.g., Refs. [25,26,27]).

In this work we analyze an effective broadness of the FPTD for a Gaussian polymer diffusion within a channel with a periodically-varying cross-section, as depicted schematically in Figure 1. In our settings here the polymer is not subject to any regular force, and hence, its center of mass performs standard unbiased diffusive motion, in contrast to the usually studied non-equilibrium driven translocation process.

To describe the polymer dynamics in such a corrugated channel, we resort to a suitably adapted Fick–Jacobs approximation, that has been generalized recently to describe dynamics of a Gaussian polymer in channels with a varying cross-section. This has been achieved by mapping the original problem onto a one-dimensional diffusion of a point-like particle in some effective potential [28].

Within such an approach, we probe the effective broadness of the FPTD by determining its coefficient of variation $\gamma =\gamma (x|x_{0})$, defined formally as
(1)$$\gamma =\sqrt{\frac{t_{2}-t_{1}^{2}}{t_{1}^{2}}}=\frac{\sigma }{t_{1}}\;,$$
where $t_{1}=t_{1}\left(x|x_{0}\right)$ and $t_{2}=t_{2}\left(x|x_{0}\right)$ are the first (MFPT) and the second moments of the FPTD, respectively, while σ denotes the standard deviation of the first-passage time. Here, the arguments *x* and $x_{0}$ signify that γ (as well as all other moments of the FPTD) depends explicitly on the parameter $x_{0}$, which determines the relative position of the reflective/absorbing boundaries with respect to the chambers and the bottlenecks of the channel, and also on the starting point $x\in (x_{0},x_{0}+L)$, where *L* is the periodicity of the channel (see Figure 1). Certainly, the approach we use here is just an approximation of realistic polymer dynamics in a realistic confinement. Nonetheless, it was proven to be quite reliable and consistent with the numerical analysis [28]. We thus expect that, although may be not quantitatively precise, our results are qualitatively meaningful and reliable in indicating the overall trends. Next, we recall that γ—the key property on which we focus here—is a very significant parameter. In the standard nomenclature, γ≪1 corresponds to situations in which the associated distribution can be considered as narrow. In this case, indeed, the MFPT can be regarded as a robust time-scale which characterizes the full FPTD exhaustively well. Conversely, when γ is comparable to (or even exceeds) 1, or in other words, when the standard deviation is comparable to (or even bigger than) the mean value, the associated distribution is regarded as broad. In this case, the first-passage events are completely defocused in time, such that one can no longer specify a unique time-scale.

We present a systematic analysis of the dependence of γ on a variety of the system’s parameters, such as the starting point, geometry of the channel—the height of the entropy barrier, right-left asymmetry or a relative position of the reflecting/absorbing boundaries with respect to the location of the chambers and the bottlenecks in a channel—as well as on the length of the polymer. We observe that the coefficient of variation γ of the FPTD can attain values which differ by orders of magnitude. The minimal values that γ achieves for a certain range of parameters never drop below 0.6, meaning that the fluctuations are rather significant. Typically, however, γ exceeds 1 and can become as large as several hundreds, in which case the fluctuations of the first-passage time are much bigger than its mean value. Overall, our analysis signifies that studying the translocation dynamics in terms of the MFPT only is rather meaningless and one has, as an actual fact, to quantify the fluctuations. Desirably, one has to know the full FPTD. Our goals are, however, more modest here, although our results clearly demonstrate the break-down of the MFPT as a unique characteristic time-scale for the first-passage events in the translocation phenomena. The analysis of the structure of the full FPTD will be presented in our future work, and here we scrutinize the dependence of γ on all the pertinent parameters.

The paper is outlined as follows: In Section 2 we formulate our model and introduce basic notations. In Section 3 we present our main results which quantify the dependence of γ on the parameters of our model system. Finally, we conclude in Section 4 with a brief recapitulation of our results and a perspective.

## 2. Model

Consider a 2D channel with a periodically-varying cross-section, (as depicted in Figure 1), whose half-thickness h(x) at position *x* is described by
(2)$$h\left(x\right)=h_{0}+h_{1}\left\{\begin{array}{cc} -\cos\left(\frac{\pi x}{L_{1}}\right) & \mathrm{mod}\left(x,L\right)<L_{1}\\ \cos\left(\pi \frac{x-L_{1}}{L-L_{1}}\right) & \mathrm{mod}\left(x,L\right)\geq L_{1} \end{array}\right.$$
where $h_{0}$ is the average cross-section of the channel, $h_{1}$ is the amplitude of the modulation, *L* is the periodicity, and the parameter $L_{1}$ controls the left–right channel symmetry.

As our analysis is focused on the first-passage properties, we imposed specific boundary conditions; a reflecting boundary condition at $x=x_{0}$, and an absorbing one at $x=x_{0}+L$. We expect, on intuitive grounds, that the form and the effective broadness of the FPTD will depend on the relative position of the absorbing and the reflecting boundary conditions (i.e., $x_{0}$ and $x_{0}+L$) with respect to the channel bottlenecks and chambers [29]. To this end, we used $x_{0}/L$ as a control parameter, which takes this (possible) geometrical effect into account.

According to the generalized Fick–Jacobs approximation (see Refs. [30,31,32] for more details), the dynamics of the center of mass of a Gaussian polymer inside a 2D channel with the varying cross-section, Equation (Equation 2), can be mapped onto the diffusive dynamics of a point-like particle in the effective potential [28]:(3)$$A\left(x\right)=\frac{1}{\beta }\ln[\frac{16h(x)}{h_{0}\pi^{2}}\sum_{p=1,3,\ldots }^{\infty }\frac{1}{p^{2}}\exp\left(-\frac{\pi^{2}p^{2}R_{G}^{2}}{4h^{2}\left(x\right)}\right)],$$
where $R_{G}=b\sqrt{N/9}$ is the 2D gyration radius of a Gaussian polymer comprising *N* identical monomers, each of linear size *b*, in an unbounded system, $\beta^{-1}=k_{B}T$ is the thermal energy with $k_{B}$ being the Boltzmann constant and *T*—the temperature, and *x* is the coordinate along the channel’s axis (see, Figure 1). We remark that similar results can be obtained for axis-symmetric 3D-channels, see Ref. [28]. Assuming overdamped Langevin dynamics, we write down the following associated Fokker–Planck equation for the evolution of the probability density function P(x,t):(4)$$\frac{\partial }{\partial t}P\left(x,t\right)=D_{N}\frac{\partial }{\partial x}\left[\beta P\left(x,t\right)\frac{\partial }{\partial x}A\left(x\right)+\frac{\partial }{\partial x}P\left(x,t\right)\right]$$
where $D_{N}$ is the diffusion coefficient of the center of mass of the polymer. From the last equation, one derives a differential equation obeyed by the MFPT, as well as a cascade of corresponding equations for all higher moments of the FPTD. The MFPT $t_{1}$ obeys [33]:(5)$$-\beta \left(\frac{\partial }{\partial x}A\left(x\right)\right)\frac{\partial t_{1}}{\partial x}+\frac{\partial^{2}t_{1}}{\partial x^{2}}=-\frac{1}{D_{N}},$$
while the second moment, $t_{2}$, can be determined by solving a bit more complicated differential equation [33]:(6)$$-\beta \left(\frac{\partial }{\partial x}A\left(x\right)\right)\frac{\partial t_{2}}{\partial x}+\frac{\partial^{2}t_{2}}{\partial x^{2}}=-\frac{2t_{1}}{D_{N}}.$$

The reflecting condition at $x=x_{0}$ and the absorbing one at $x=x_{0}+L$ entail the following boundary conditions:(7)$$\begin{array}{cc} \frac{\partial }{\partial x}t_{1,2}{|}_{x=x_{0}} & =0\\ t_{1,2}{|}_{x=x_{0}+L} & =0. \end{array}$$

Differential Equations (Equation 5) and (Equation 6), subject to the boundary conditions in Equation (Equation 7), can be solved analytically by standard means for an arbitrary effective potential A(x), Equation (Equation 3). This gives the following explicit expressions for the first two moments of the FPTD,
(8)$$\begin{array}{cc} t_{1}= & \frac{1}{D_{N}}\int_{x_{0}}^{x_{0}+L}dx^{\prime }e^{\beta A(x^{\prime })}\int_{x_{0}}^{x^{\prime }}dx^{\prime \prime }e^{-\beta A(x^{\prime \prime })}-\frac{1}{D_{N}}\int_{x_{0}}^{x}dx^{\prime }e^{\beta A(x^{\prime })}\int_{x_{0}}^{x^{\prime }}dx^{\prime \prime }e^{-\beta A(x^{\prime \prime })} \end{array}$$
(9)$$\begin{array}{cc} t_{2}= & \frac{2}{D_{N}}\int_{x_{0}}^{x_{0}+L}dx^{\prime }e^{\beta A(x^{\prime })}\int_{x_{0}}^{x^{\prime }}dx^{\prime \prime }t_{1}\left(x^{\prime \prime }\right)e^{-\beta A(x^{\prime \prime })}-\frac{2}{D_{N}}\int_{x_{0}}^{x}dx^{\prime }e^{\beta A(x^{\prime })}\int_{x_{0}}^{x^{\prime }}dx^{\prime \prime }t_{1}\left(x^{\prime \prime }\right)e^{-\beta A(x^{\prime \prime })}, \end{array}$$
which are valid for an arbitrary form of the effective potential A(x). Note, however, that the integrals over an exponentiated potential A(x) cannot be performed in an explicit form and hence, we resorted to a numerical analysis. In Section 3 below we present the results based on the numerical integration of the expressions in Equations (Equation 8) and (Equation 9) using standard Mathematica packages.

We close this Section with the following remark. For a constant cross-section channel (such that h(x)=const and hence, $\partial_{x}A\left(x\right)=0$) the integrals in Equations (Equation 5) and (Equation 6) can be performed exactly. Indeed, in this case one deals simply with a standard diffusion on a bounded one-dimensional interval (0,L), with a reflecting boundary placed at x=0 and an adsorbing one at x=L. This is an exactly solvable and well-studied model (see, e.g., Refs. [10,11,17] and references therein). Without any loss of generality, we set $x_{0}=0$ and obtain from Equations (Equation 5) and (Equation 6) the following well-known results (see, e.g., Ref. [17] presenting the moments of the FPTD of arbitrary order explicitly in form of the Euler polynomials)
(10)$$\begin{array}{c} t_{1}^{0}=\frac{L^{2}-x^{2}}{2D_{N}}\;,t_{2}^{0}=\frac{1}{2D_{N}^{2}}\left[x^{2}\left(\frac{x^{2}}{6}-L^{2}\right)+\frac{5}{6}L^{4}\right]\;, \end{array}$$
which, in turn, provide an explicit, closed-form expression for the coefficient of variation γ, Equation (Equation 1). Note that γ is independent of $D_{N}$ and hence, of *N*. We observe that here γ attains its minimum at x=0 (i.e., when the center of mass of a polymer is right at the reflecting boundary at t=0), and this minimal value is given by $\gamma =\sqrt{2/3}\approx 0.82$. Consequently, even in this somewhat trivial case the minimal standard deviation of the first-passage time is only slightly smaller than the MFPT, i.e., the fluctuations are nonetheless sufficiently large and cannot be safely discarded. More striking, for *x* close to *L*, (i.e., when *x* is close to the adsorbing boundary), $\gamma ∼1/\sqrt{\left(L-x\right)}$ in the leading order in (L−x), implying that the standard deviation can be much larger than the MFPT. Here, of course, the MFPT alone does not describe exhaustively well the statistics of the first-passage times—the spread of fluctuations is much larger than the mean value, such that the first-passage times are defocussed. We note that we expect the same behavior of γ as a function of *x* for our model with a translocating polymer: Recall that within the generalized Fick–Jacobs approach the dynamics of the center of mass of a polymer in a varying cross-section channel are reduced to a standard diffusion in the presence of an effective potential. As a consequence, the basic features should be essentially the same, but the overall behavior should turn out to be richer since many other physical parameters come into play due to the effective potential.

## 3. Results

### 3.1. x-Dependence

The first issue we focused on is the dependence of the coefficient of variation γ of the FPTD on the relative distance of the starting point *x* from the location of the reflecting boundary, placed at $x_{0}$. To this end, in Figure 2 we depict γ for a polymer with a fixed number of beads N=90, as a function of $(x-x_{0})/L$ for different values of the control parameter $x_{0}/L$. We noticed that γ is the smallest and is close to 1 when the starting point *x* of the polymer center of mass is close to the reflecting boundary. In this case, γ was very weakly dependent on the actual value of the control parameter $x_{0}/L$. For $x_{0}/L=1$, the coefficient of variation was almost constant (approximately equal to 1) for a wide range of variation of $(x-x_{0})/L$. It started to grow with $(x-x_{0})/L$ and attained a maximal value, which is slightly below 10, when $(x-x_{0})/L$ approached its maximal value 1. The situation turned to be more interesting for intermediate values of the control parameter, (i.e., when the reflecting boundary is within a broad chamber). For such values of $x_{0}/L$, the coefficient of variation became a non-monotonic function of $(x-x_{0})/L$; upon a gradual increase of $(x-x_{0})/L$, γ first grew until it reached some peak value, then it decreased, stayed almost constant in some region of values of the variable $(x-x_{0})/L$, and then, for $(x-x_{0})/L$ being close to 1, γ started to grow again reaching a peak value of order 10 for $(x-x_{0})/L=1$. Lastly, for the smallest values of the parameter $x_{0}/L$, (such that the reflecting boundary is located sufficiently close to the center of the first bottleneck), we observed a different type of behavior; γ was of order of 1 for an extended range of values of *x*, then for $(x-x_{0})/L\approx 0.5,0.7$ and 0.9 (for $x_{0}/L=0,0.1$ and 0.3, respectively) we observed an abrupt growth of γ to very big values, exceeding $10^{2}$. We concluded that γ, as could be expected on intuitive grounds (see also our remark at the end of Section 2), increased with increasing *x*, i.e., fluctuations became more and more relevant while approaching the absorbing boundary condition. Overall, γ varies very strongly, being at least of order of 1 but may also reach very big values. This means that the FPTD can become effectively very broad. As a consequence, fluctuations of the first-passage time can never be safely discarded.

Another intriguing aspect of the behavior depicted in Figure 2 is the apparent focusing/defocusing effect of a periodic corrugation. Comparing the behavior of γ for a constant cross-section channel (red dashed curve), we observed for $x_{0}/L≃0,0.9$ the values of γ for varying-section channels are significantly less than those obtained for a constant-section channel, which means that in this case the periodic corrugation reduced fluctuations leading to a certain focusing of random first-passage time around its mean value. In contrast, for $x_{0}/L≃0.5-0.8$, the corrugation of the channel entailed larger values of γ, as compared to a constant cross-section case, meaning that the first-passage times became more defocused—the amplitude of fluctuations was enhanced.

### 3.2. $x_{0}$-Dependence

We concentrated next on the dependence of γ on $x_{0}/L$ for two particular values of the starting point *x*; $x=x_{0}$ and $x=x_{0}+L/2$. Figure 3 presents such a dependence for different values of *N* (with lighter colors corresponding to bigger values of *N*). We observed that for $x=x_{0}$, i.e., when the starting point was located exactly at the reflecting boundary, γ, (which was close to 1 for $x_{0}=0$), dropped to some minimal value slightly below 0.7 at $x_{0}\approx 0.8$ and then increased for larger values $x_{0}/L$. The rate at which γ first dropped and then increased again depended on *N*, such that the effective width of the deep region was smaller for longer polymers than for the shorter ones. For $x_{0}/L$ close to 1, γ decreased again. The trend was completely inverted for $x=x_{0}+L/2$, when the starting point was right in the middle between the reflecting and the absorbing boundaries. Here, γ stayed constant (of order of 1) within an extended region of variation of the parameter $x_{0}/L$. This constant value was only very weakly dependent on the polymer’s length. Upon a further increase of $x_{0}/L$, γ raised abruptly to a peak value which now strongly depended on *N*. In particular, for N=90, one had γ≈50, while for a smaller value, N=50, the coefficient of variation γ≈10. For bigger values of $x_{0}/L$, γ decreased with the growth of $x_{0}/L$ and saturated at a constant value, which was seemingly independent of *N*. Interestingly enough, the effective width of the hump region in panel Figure 3b also shows an opposite trend as compared to the width of the deep in panel Figure 3a; the hump region becomes essentially wider and more pronounced for longer polymers than for shorter ones.

### 3.3. N-Dependence

In Figure 4 we focused specifically on the *N*-dependence of the coefficient of variation of the FPTD. Panel Figure 4a presents such a dependence for a polymer whose starting point is exactly at the location of the reflecting boundary, i.e. $x=x_{0}$. We observed that γ was a non-monotonic function of the polymer size *N*. For $x_{0}/L≃0.85$, i.e., when the location of the reflecting boundary was sufficiently close to the center of the first bottleneck, γ first decreased with *N*, attained a minimal value for some $N=N^{*}$, and then increased, again approaching 1. For $x_{0}/L=0.9$ and bigger, i.e., when the location of the reflecting boundary was displaced towards the center of the broad chamber, the behavior of γ was even more complicated. Here, the coefficient of variation first grew with *N*, attained some local maximal value, then decreased approaching a minimum, and then started to grow again. Such behavior is clearly seen in the case $x_{0}/L=0.9$, but we expected that it was a generic feature and should persist for larger values of $x_{0}/L$, if we consider also values of *N* bigger than $10^{2}$. Note, as well, that the coefficient of variation can be bigger (defocusing) or smaller (focusing) than the value of γ specific for the constant cross-section channels (red dashed line in Figure 4), showing that the geometry of the channel can lead to a larger or a smaller spread of fluctuations of the first-passage time around its mean value.

Panel b in Figure 4 shows an analogous dependence in the case when the starting point of the polymer $x=x_{0}+L/2$. Here, γ appeared to be a monotonic function of *N*, but whether it was a monotonically decreasing or a monotonically increasing function depended essentially on the value of the control parameter $x_{0}/L$. Namely, when the reflecting boundary was sufficiently close to the center of the first bottleneck, γ appeared to be a monotonically decreasing function and approached (from above) the value of 1. In contrast, when the reflecting boundary was displaced towards the broad chamber, γ was a monotonically increasing function and exhibited an unbounded growth with *N*. Overall, the results presented in Figure 4 imply that the coefficient of variation strongly depends on the polymer length, in a sharp contrast to the polymer translocation time across a straight pore, for which γ does not depend on *N* [34].

### 3.4. Dependence on the Entropic Barrier Δ S

So far we have dealt with a fixed amplitude of the corrugation and a prescribed shape. It seems interesting to probe, as well, the effect of different amplitudes of the corrugation on the coefficient of variation of the FPTD. To this end, we analyzed the dependence of γ on the dimensionless entropic barrier $\Delta S=\log[\left(h_{0}+h_{1}\right)/\left(h_{0}-h_{1}\right)]$. This dependence is depicted in Figure 5.

Figure 5a shows that for small values of $x_{0}$, ($x_{0}/L=0,0.1$ and 0.2), the coefficient of variation was a monotonically increasing function of ΔS and γ approached 1 from below when ΔS→∞. On contrary, for bigger values of the control parameter, the coefficient of variation of the FPTD developed a local minimum whose depth increased with ΔS for $x_{0}/L≃0.8$. Further increase of $x_{0}$ would smear out the non-monotonous behavior. Figure 5b shows in detail the crossover between monotonous and non-monotonous behavior. Upon increasing $x_{0}$, the depth and the width of the minimum of γ as function of ΔS decreased and the position of the minimum was shifted towards larger values of ΔS. Eventually, for even larger values of $x_{0}/L$ a monotonous behavior was restored. Interestingly, as one may infer from Figure 5b, the value of the minimum was below the one which corresponds to that of a constant cross-section channel. Hence, upon some fine-tuning of the shape of the channel, it was possible to reduce the magnitude of fluctuations of the first-passage time which offered a possibility of a geometry control. Consequently, in line with our earlier discussion, the presence of the entropic barrier can reduce the coefficient of variation of the FPTD. We noted that this effect was due to the fact that the first, $t_{1}$, and the second, $t_{2}$, moments of the FPTD depended differently on the value of the entropic barrier, ΔS. Hence, Figure 5 suggest that it is possible to attain a regime in which $t_{1}$ has a steeper dependence on Δs than $t_{2}$ by the fine-tuning of the parameters. This is quite surprising, however, because it is generally expected that the presence of inhomogeneities and barriers enhances the magnitude of fluctuations.

### 3.5. $L_{1}$-Dependence

Finally, we analyzed the dependence of γ on the left–right asymmetry of the channel, i.e., when $L_{1}\neq L/2$ (see Equation (Equation 1)). Figure 6a shows that the dependence of γ on $x_{0}$ was quite robust with respect to a change of the value of $L_{1}$—the main conclusion being that the position of the minimum was shifted towards smaller values of $x_{0}$. More interesting was the dependence of γ on $L_{1}$ itself. Figure 6b shows that this dependence was non-monotonous, in general, and hence, there existed an optimal left–right asymmetry for which γ attained its minimal value. Therefore, in line with our earlier discussion, it appears that the corrugation of the channel can result in a more focused behavior of the first-passage times.

## 4. Conclusions

To conclude, we studied here the first-passage statistics of a Gaussian polymer translocation process in a periodically-corrugated channel within the framework of a generalized Fick–Jacobs approach [28]. Employing this approximation, we derived closed-form, explicit expressions for the first (mean first-passage time, MFPT) and for the second moments of the first-passage time distribution (FPTD), which permitted us to probe the effective broadness of the latter by analyzing the behavior of the so-called coefficient of variation γ of the FPTD, defined as the ratio of the standard deviation and of the MFPT. Numerically integrating the above expressions, we presented a systematic analysis of γ as function of a variety of the system’s parameters—the polymer length, the position of the starting point, the relative position of a target point with respect to the bottlenecks and chambers of the channels, the value of the entropy barrier, and the left–right asymmetry of the channels.

We showed that γ exhibits quite a rich behavior. Curiously enough, in some cases γ appears to be a non-monotonic function with two minima or two maxima, and shows a very big variation (over several decades) upon a slight change of one of the parameters. In other situations, conversely, it can be a monotonic function of its parameters. For some values of the starting point and the relative positions of the channel’s bottlenecks and chambers with respect to the precise location of the reflecting and absorbing boundaries, γ becomes smaller than its counterpart for the constant thickness channels, meaning that the channel’s geometry entails some focusing of the first-passage times around its mean values. For other values of the system’s parameters, on the contrary, the geometry of the channels leads to the defocusing of the first-passage times, i.e., a larger spread of fluctuations around the MFPT.

An important general observation is that γ never significantly drops below 1 and, in fact, can attain very large values. This means, in turn, that the first-passage times in the process of a polymer translocation are typically very defocused. Even for γ slightly below 1, the fluctuations around the mean are of the same order as the mean value itself. For γ substantially exceeding 1, which is very often the case, the fluctuations are much larger than the mean value (the MFPT). This implies that the MFPT alone cannot characterize the first-passage statistics of the translocation process exhaustively well. This circumstance is crucial for experimental analyses that typically rely on small statistical samples.

In this regard, a meaningful continuation of our present analysis is to go beyond the first two moments and to concentrate on the full FPTD. Another perspective problem is to include a constant force acting on the polymer and pointing along the corrugated channel, as happens in many pertinent realistic systems. These are two main directions of our further research.

## Figures and Tables

**Figure 1 polymers-11-00251-f001:**
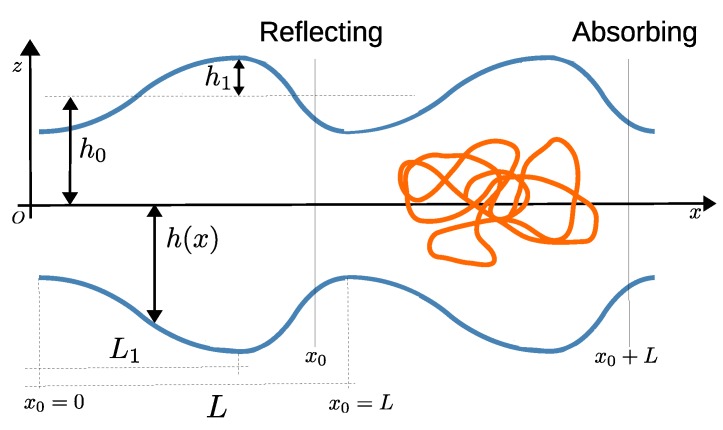
A polymer (orange curve) in a channel with a periodically-varying cross-section and impermeable walls (blue solid curves). The bottlenecks of the channel have half-thickness $h_{0}-h_{1}$, (see Equation (Equation 2)), while the half-thickness of broad chambers is equal to $h_{0}+h_{1}$. The first bottleneck on the left-hand-side of the channels is fixed to be located at the origin *O*, i.e., x=0. The vertical lines $x_{0}$ and $x_{0}+L$ indicate the positions of the reflecting and absorbing boundaries, respectively. Varying $x_{0}$ at a fixed profile h(x), we quantify the effect of a relative position of the absorbing/reflecting boundaries with respect to the bottlenecks and chambers of the channel onto the effective broadness of the first-passage time distribution (FPTD), as probed here by the value of γ.

**Figure 2 polymers-11-00251-f002:**
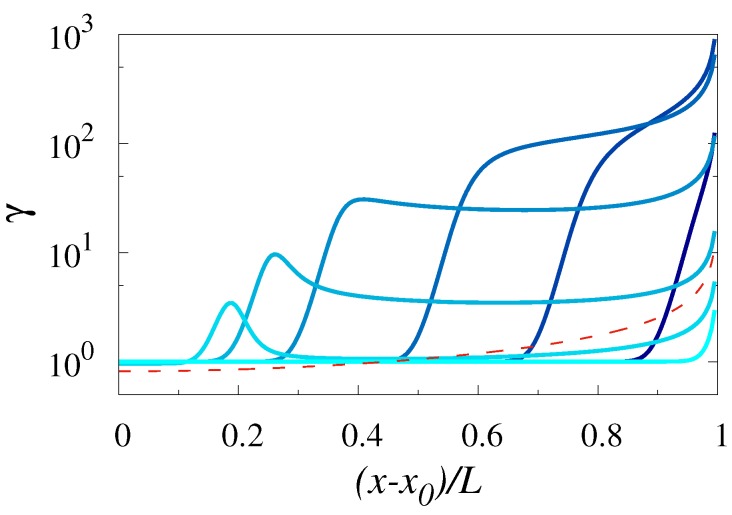
Coefficient of variation γ of the FPTD for a Gaussian polymer translocating across a varying-section, left–right symmetric ($L_{1}=L/2$) channel, as a function of $(x-x_{0})/L$. The channel geometry was characterized by $h_{0}/L=0.25$ and $h_{1}/h_{0}=0.8$ (see Equation (Equation 2)). The number *N* of beads in a polymer was fixed, N=90. The control parameter, $x_{0}/L$, assumes values of 0.1,0.3,0.5,0.7,0.8,0.85 and 1, with lighter colors corresponding to larger values of $x_{0}/L$. The dashed line indicates the coefficient of variation of the FPTD in a constant cross-section channel.

**Figure 3 polymers-11-00251-f003:**
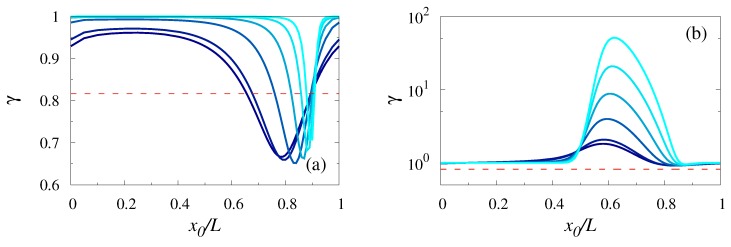
Coefficient of variation γ of the FPTD as a function of $x_{0}/L$ for $x=x_{0}$ (panel **a**) and $x=x_{0}+L/2$ (panel **b**) for N=5,10,30,50,70, and 90 (with lighter colors corresponding to larger values of *N*) with $L_{1}=L/2$ and $h_{0}/L=0.25$ and $h_{1}/h_{0}=0.8$. The channel geometry is the same as in Figure 2. The horizontal dashed line in both panels represents the value of γ for a constant section channel (recall that for this case γ was independent of the polymer length *N*).

**Figure 4 polymers-11-00251-f004:**
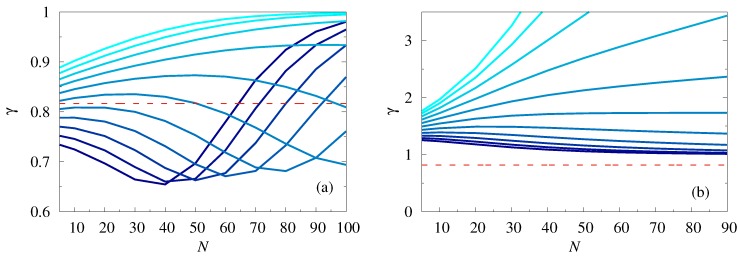
Coefficient of variation γ of the FPTD as a function of *N* for $x=x_{0}$ (panel **a**) for $x_{0}/L$ ranging from 0.85 to 0.95 with steps of 0.01, and $x=x_{0}+L/2$ (panel **b**) for $x_{0}/L$ ranging from 0.45 to 0.55 with steps of 0.01 (lighter colors stand for larger values of $x_{0}/L$) with $L_{1}=L/2$ and $h_{0}/L=0.25$ and $h_{1}/h_{0}=0.8$. The horizontal dashed line in the panels (**a**) and (**b**) represents the value of γ for $x=x_{0}$ for a constant section channel (recall that in this case γ is independent of the polymer length *N*).

**Figure 5 polymers-11-00251-f005:**
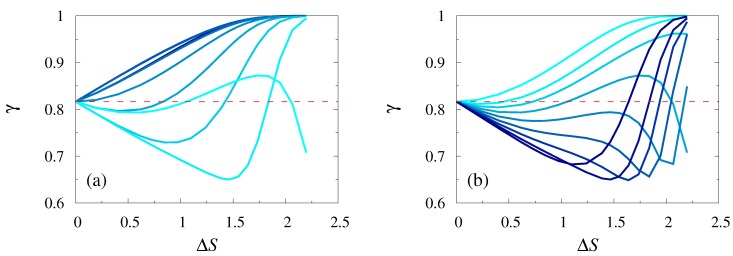
Coefficient of variation γ as a function of the reduced entropic barrier $\Delta S=\log[\left(h_{0}+h_{1}\right)/\left(h_{0}-h_{1}\right)]$ with $L_{1}=L/2$. The polymer length was fixed, N=90, and $x_{0}/L$ is ranging from 0 to 0.9 with steps of 0.1 (panel **a**) with lighter colors corresponding to larger values of $x_{0}/L$. In panel (**b**) N=90 and $x_{0}/L=0.75,0.8,0.825,0.85,0.875,0.9,0.925,0.95,1$, respectively. The horizontal dashed line in panels (**a**) and (**b**) represents the value of γ for a constant section channel (for this case γ is independent of the polymer length *N*)

**Figure 6 polymers-11-00251-f006:**
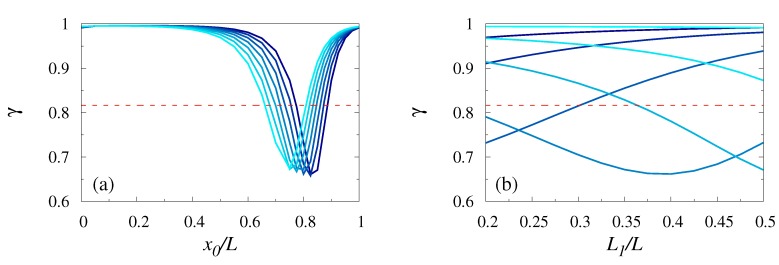
The coefficient of variation of the FPTD for left/right asymmetric channels with $h_{0}/L=0.25$ and $h_{1}/h_{0}=0.8$. Panel (**a**): Coefficient of variation γ as function of $x_{0}/L$ for $L_{1}=0.2,0.25,0.3,0.35,0.4,0.45,0.5$ (lighter colors correspond to smaller values of $L_{1}$) for ΔS=1.7 and N=90. Panel (**b**): Coefficient of variation γ as a function of $L_{1}$ for ΔS=1.7, N=90 and $x_{0}/L=0.5,0.6,0.7,0.8,0.9,1$ (lighter colors correspond to larger values of $x_{0}/L$) and. The horizontal dashed line in both panels represents the value of γ for a constant section channel (for this case γ is independent of the polymer length *N*).

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
