# Peer review of "Polymer Translocation Across a Corrugated Channel: Fick–Jacobs Approximation Extended Beyond the Mean First-Passage Time"

_polymers, 2019, doi:10.3390/polym11020251_

Round 1

Reviewer 1 Report

In this work, the authors studied the effective broadness of the first passage time distribution for Gaussian chains translocating across a corrugated channel. The broadness, denoted by gamma, was quantified by the ratio of the standard deviation and the mean first passage time. The translocation problem was first mapped into a 1D diffusion problem of a point-like particle in an effective potential under the Fick-Jacobs approximation. The first moment and the second moment of the first passage time were then determined by solving numerically the two differential equations derived from the associated Fokker-Planck equation under the reflecting and absorbing boundaries. The coefficient of variation (gamma) was then studied by varying the positions of the boundaries in the channel, the starting position of the polymer, the chain length, and the amplitude of the corrugation and the left-right asymmetry of the channel. The main conclusion that the authors addressed is that the fluctuation of the first passage time is so large that the translocation cannot be uniquely characterized by the mean first passage time.

The topics of this work are quite interesting and have its scientific value. The presentation is generally clear. However, the correctness of the calculation is hard to be justified in the current text. The authors failed to give clear physical pictures in associated with the obtained results, to explain where and why the large fluctuations come and how the chains evolve across the corrugated channel in these conditions. Particularly, the calculation showed that the fluctuation can be as large as hundred or thousand times of the mean first passage time. It is hard to believe it. Without appropriate physical explanations or proofs, we do not know in which degree the numerical calculations were valid and how far we could trust the results. Therefore, I do not recommend the manuscript for publication in the current form. A major revision is needed. The authors should provide suitable physical explanations, focusing on the reason for the occurrence of the large fluctuations and describing the possible dynamics of chain associated with the fluctuations so that the readers could have a vision what happened exactly behind the large gamma. It will increase significantly the value of the paper.

Followings are other points that the authors should also resolve:

1.      The origin O and the asymmetry parameter L1 should be indicated in Fig.1 to help readers to understand easily the model.

2.      The definition of beta (inverse thermal energy) is not given in Eq.(3).

3.      The authors should mention by which numerical method they integrated the expressions Eq. (8) and (9). Also how did they know for sure that the numerical results were reliable?

4.      The authors should divide the “Results” section into subsections to increase the readability of the manuscript. For example, Fig.2 presented the results concerning the varying of the starting point of translocation for different choices of the positions of the boundaries. Fig.3 showed the effect of the boundary positions for different chain lengths at two different starting positions, and so on. These analyses are very good and systematic. It is pity that the authors did not present the results in an organized way. Through a suitable subdivision of the section with appropriate subsection titles, the logical flow and the value of the study can be clearly presented.

5.      Some of the control parameter (X0/L) described in the text cannot be found in the associated figures. For example, the control parameter 0 is missing in Fig. 2, and so forth.

6.      In addition to providing physical explanations, it will be great if the authors could make some links or give comments with/on experiments.

Author Response

The topics of this work are quite interesting and have its scientific value. The presentation is generally clear. However, the correctness of the calculation is hard to be justified in the current text. The authors failed to give clear physical pictures in associated with the obtained results, to explain where and why the large fluctuations come and how the chains evolve across the corrugated channel in these conditions. Particularly, the calculation showed that the fluctuation can be as large as hundred or thousand times of the mean first passage time. It is hard to believe it. Without appropriate physical explanations or proofs, we do not know in which degree the numerical calculations were valid and how far we could trust the results.

First, we are a bit puzzled by the remark of the Referee that it is hard to believe that  the standard deviation can be thousand times larger than the mean value. In fact, it is a common knowledge that for diffusive processes starting close to the target the FPTD distribution becomes very broad and the coefficient of variation of the FPTD attains very large values (diverges). In this case, of course, the mean first passage time is not a robust measure. A thorough discussion of this point can be found in References 17-21 of our reference list.  

There is no deep physics behind such a behavior: namely, it occurs due to a typical behavior of the searching trajectories - for a diffusive process starting close to the adsorbing point there are a few, rather direct trajectories arriving to the adsorbing point within a very short time, while the majority of trajectories goes away and samples the entire volume before eventually returning to the target. This causes a pronounced disparity in first passage times, and fluctuations around the mean value are by and large much bigger than the mean value.  The full FPTD in this case has a very pronounced structure - an exponential left tail rising to the peak value defining the most probable first passage time, a power-law descent from the peak value (Smirnov-L\'evy form), which is eventually tempered, at time approximately equal to the mean-first-passage time, by an exponential right-tail of the distribution. The difference between the most probable and the mean-first-passage time can span many orders of magnitude, which implies that the first passage times are very defocused.

There is absolutely no surprise that we observe such a behavior in the model under study. Recall that, in essence, the generalized Fick-Jacobs approach maps the dynamics of the center of mass of a Gaussian polymer in a corrugated channel, onto a Brownian motion in an appropriately defined effective potential. As a consequence, one can generally expect that some basic features specific to Brownian motion in absence of an external potential, (including large values of gamma for a starting point close to the target), will also persist for the model under study.  Therefore, the predicted values of gamma, resulting
from numerical evaluation of the integrals in the exact expressions  (8) and (9) using standard Mathematica packages,  
are physically correct and not counter-intuitive.   

On the other hand, we admit that these points, without a preliminary discussion,  may seem to be surprising to
 a non-expert in the field of first-passage phenomena. To this end, we have added several sentences into the Introduction on pg. 2 highlighting the role of fluctuations, and also added a remark at the end of Sec. 2 on the divergence of the coefficient of variation for standard diffusive motion starting close to the adsorbing boundary. We believe that now the large values  of gamma will not appear as a surprise to the readers.

Other comments:
1-  The origin O and the asymmetry parameter L1 should be indicated in Fig.1 to help readers to understand easily the model.

We have amended Fig. 1 accordingly and also introduced some changes into the Figure caption.}

2- The definition of beta (inverse thermal energy) is not given in Eq.(3).

The definition of beta is introduced into the text below Eq. (3).}

3- The authors should mention by which numerical method they integrated the expressions Eq. (8) and (9). Also how did they know for sure that the numerical results were reliable?

Equations have been numerically integrated with standard \textit{Mathematica} that provides by default a working precision of 16 digits. A corresponding remark is introduced into the text after Eqs.(8) and (9).

4- The authors should divide the Results section into subsections to increase the readability of the manuscript. For example, Fig.2 presented the results concerning the varying of the starting point of translocation for different choices of the positions of the boundaries. Fig.3 showed the effect of the boundary positions for different chain lengths at two different starting positions, and so on. These analyses are very good and systematic. It is pity that the authors did not present the results in an organized way. Through a suitable subdivision of the section with appropriate subsection titles, the logical flow and the value of the study can be clearly presented.

We thank the Referee for this suggestion and have accordingly structured
the section devoted to the presentation of our Results.

5- Some of the control parameter (X0/L) described in the text cannot be found in the associated figures. For example, the control parameter 0 is missing in Fig. 2, and so forth.

We thank the Reviewer for this remark. We have fixed these points in the new version of the manuscript.

6- In addition to providing physical explanations, it will be great if the authors could make some links or give comments with/on experiments.

We believe that we have already commented on this topic in the introduction of our manuscript. However, in order to reinforce this point, we have added a final remark in the conclusions. We also believe that our predictions, as they stand, can be accessed experimentally and our work will prompt an interest of experimentalists.

Reviewer 2 Report

This manuscript reports on a theoretical analysis of polymer translocation dynamics in a corrugated channel.  The authors evaluated the first passage time distribution (FPTD) through calculating the so-called coefficient of variation within a framework of a generalized Fick-Jacobs approach.  Their results indicated that FPTD is generally broad irrespective of the channel geometries and entropic barriers involved whereby cautioned not to characterize the translocation statistics by the mean first passage time alone.

This is a well-written paper providing useful information for experimental researcher who study polymer translocation in nanopores and nanochannels.  I will recommend publication of this nice work after the authors address the following points:

1.     As remarked by the authors, it sounds surprising that larger entropic barrier leads to less variation in the FRTD.  It is better explained in more detail what can be a possible physics underlying the intriguing behavior.

2.     Similar to the above point, the data in Fig. 6 showing contributions of channel symmetry should be also discussed in more detail.

3.     In Fig. 4 caption, x_0/L was stated to range from 0.85 to 0.95 and 0.45 to 0.55 in panels a and b, respectively.  Meanwhile, page 7 discusses the dependence of γ in a wider range of x_0/L.  For example, γ at x_0/L = 0 to 0.2 is mentioned in the main text without any data provided.  It is preferable for readers to see the data being discussed.

4.     There are several typos: “FPDT” in page 3 should be “FPTD”; “non-monotoneous” in page 8 should be “non-monotonous”; “which corresponds a constant …” in the same page should be “which corresponds to that of a constant …”.

5.     Symbols in the figures should be italicized.

Author Response

1- As remarked by the authors, it sounds surprising that larger entropic barrier leads to less variation in the FPTD.  It is better explained in more detail what can be a possible physics underlying the intriguing behavior.

 We thank the Reviewer for pointing on this issue. We have addressed it in the new version of the manuscript.

2- Similar to the above point, the data in Fig. 6 showing contributions of channel symmetry should be also discussed in more detail.

We have expanded the comments on Fig. 6.

3- In Fig. 4 caption, x_0/L was stated to range from 0.85 to 0.95 and 0.45 to 0.55 in panels a and b, respectively.  Meanwhile, page 7 discusses the dependence of \gamma in a wider range of x_0/L.  For example, \gamma at x_0/L = 0 to 0.2 is mentioned in the main text without any data provided.  It is preferable for readers to see the data being discussed.

We thank the Reviewer for pointing on this issue. We have rephrased the text so to refer just to the data in the figure.

4- There are several typos: “FPDT” in page 3 should be “FPTD”; “non-monotoneous” in page 8 should be “non-monotonous”; “which corresponds a constant '' in the same page should be “which corresponds to that of a constant ”

 We have corrected these typos.

5- Symbols in the figures should be italicized.

We have changed the symbols in the figures.

Round 2

Reviewer 1 Report

The authors have made efforts to revise the manuscript. I will recommend the paper for publication if the authors could fix or clarify the following issues.

(1)   Please check again the Figure 1. Are the indications, x0=0 and x0=L, in the figure correct?

Also in the caption, h0-h1 and h0+h1 should be the “half”-thickness.

(2)   Why did the authors call h(x) the half-thickness (page 4)? Why didn’t they call it the radius of the channel? Were the results obtained from the 2D effective potential or from the 3D effective potential (Eq. 3)? The authors forgot to say it clearly in the text.

(3)   The author should discuss if the differences were important for the results obtained from the 2D and the 3D effective potentials.

Author Response

We the Reviewer for appreciating our amendments. In the following we reply in detail to the comments of the Reviewer:

    comment:   Please check again the Figure 1. Are the indications, x0=0 and x0=L, in the figure correct?

We have checked and indeed the label are properly located. Moreover they are in agreement with Eq.2.

comment: Also in the caption, h0-h1 and h0+h1 should be the “half”-thickness.

We thank the Reviewer for identifying this issue. We have change the caption accordingly

comment: Why  did the authors call h(x) the half-thickness (page 4)? Why didn’t they  call it the radius of the channel? Were the results obtained from the 2D  effective potential or from the 3D effective potential (Eq. 3)? The  authors forgot to say it clearly in the text. The  author should discuss if the differences were important for the results  obtained from the 2D and the 3D effective potentials.

We have specified in the new version of the manuscript that our results indeed are for 2D channels and we have commented on possible differences with axis-symmetric 3D channels.